# SAMPLE, DON'T SEARCH: RETHINKING TEST-TIME ALIGNMENT FOR LANGUAGE MODELS

## ABSTRACT

Increasing test-time computation has emerged as a promising direction for improving language model performance, particularly in scenarios where model finetuning is impractical or impossible due to computational constraints or private model weights. However, existing test-time search methods using a reward model (RM) often degrade in quality as compute scales, due to the over-optimization of what are inherently imperfect reward proxies. We introduce QALIGN, a new test-time alignment approach. As we scale test-time compute, QALIGN converges to sampling from the optimal aligned distribution for each prompt. By adopting recent advances in Markov chain Monte Carlo for text generation, our method enables better-aligned outputs without modifying the underlying model or even requiring logit access. We demonstrate the effectiveness of QALIGN on mathematical reasoning benchmarks (GSM8K and GSM-Symbolic) using a task-specific RM, showing consistent improvements over existing test-time compute methods like best-of-$n$ and majority voting. When applied with more realistic RMs trained on the TÜLU 3 preference dataset, QALIGN outperforms direct preference optimization (DPO), best-of-$n$, majority voting, and weighted majority voting on a diverse range of datasets (GSM8K, MATH500, IFEval, MMLU-Redux, and TruthfulQA). A practical solution to aligning language models at test time using additional computation without degradation, our approach expands the limits of the capability that can be obtained from off-the-shelf language models without further training.

## 1 INTRODUCTION

Language models (LMs) have demonstrated remarkable capabilities through learning from human preferences (Stiennon et al., 2022; Fernandes et al., 2023; Kaufmann et al., 2023; Peters & Schaal, 2007; Peng et al., 2019; Korbak et al., 2022b;a; Go et al., 2023). However, there are a number of problems with deploying a single aligned model: alignment approaches typically average multiple human preferences, constructing a monolithic preference model and target policy. Furthermore, approaches to adapting these models through finetuning have become increasingly impractical, requiring enormous computational resources. They are entirely impossible when model weights are private, as with many state-of-the-art models (OpenAI, 2024; Anthropic, 2024; Gemini, 2024).

While conventional alignment methods optimize a single model for aggregate performance across a distribution of prompts and then produce a random generation during inference, researchers have found great improvement from scaling the amount of compute expended at test time for each prompt (Brown et al., 2024; Snell et al., 2024). Here, multiple outputs are generated and then used to produce a final answer either via reward maximization (known as "best-of-$n$," or BoN; Gao et al., 2022; Stiennon et al., 2020; Fernandes et al., 2022), majority voting (MV; Wang et al., 2023b) or weighted majority voting (WMV; Li et al., 2023). While promising, existing search-based methods face fundamental limitations (Liu et al., 2024; Wu et al., 2024; Zhang et al., 2023; Xie et al., 2023): as inference compute scales, these methods over-optimize learned reward models (RMs), which are inherently imperfect proxies (Gao et al., 2022).

In this paper, we introduce QALIGN, a test-time alignment method that converges to sampling from the **optimal aligned distribution** (defined in §1 as the true target for existing RLHF methods) for

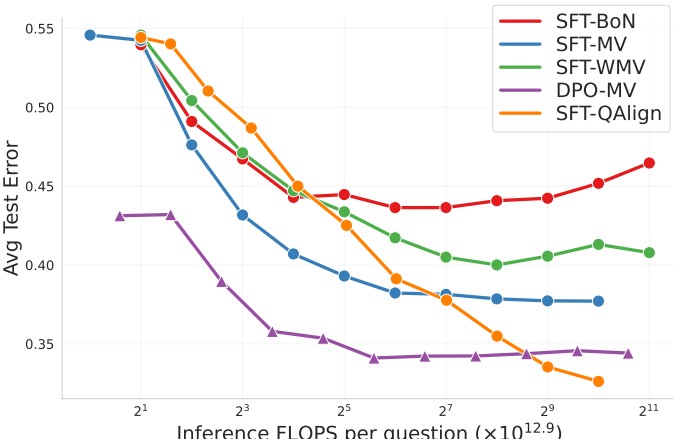

Figure 1: Average error rate across multiple evaluation datasets (GSM8K, MATH500, MMLU-Redux, TruthfulQA, and IFEval) as a function of inference-time floating point operations (FLOPS) in log scale. We compare ●**QALIGN with TÜLU3-8B-SFT** against four baselines: ▲ **majority vote (MV) TÜLU3-8B-DPO**, and applied to TÜLU3-8B-SFT the methods ● **best-of-$n$ (BoN)**, ● **MV**, and ● **weighted MV (WMV)** . All experiments use temperature 1.0 with reasoning included in model outputs. The TÜLU3-8B-DPO model results from preference finetuning TÜLU3-8B-SFT (approximately $1.75 \times 10^{19}$ FLOPs). The costs of this process are not accounted for in this plot.

each prompt, individually, as the test-time compute budget is increased. This enables more aligned responses without requiring access to the logits or training any underlying LM (only sampling from it). Specifically, we adapt QUEST (Faria et al., 2024). QUEST was designed to repurpose a language model, creating a Markov chain converging toward a (Gibbs) distribution defined by machine translation quality estimation metrics. In this work, we show that we can use QUEST to align LMs at test time according to an RM learned from preference data (in place of the quality estimation metric) and to provide a single good final prediction.

In our experiments (§4), we compare QALIGN with existing test-time compute methods on mathematical reasoning benchmarks (**GSM8K**, Cobbe et al., 2021, and **GSM-Symbolic**, Mirzadeh et al., 2024), by training a task-specific RM (§4.1), and using it to increase the capability of LLAMA3.1-8B-INSTRUCT (Llama, 2024). We obtain a consistent reduction in error as we scale inference time computation across scales, as shown in Figure 2. Unlike BoN, QALIGN's performance does not degrade as we increase the compute per question. Furthermore, when applied with more realistic RMs (§ 4.2), trained on the TÜLU3 (Lambert et al., 2024) preference dataset, where over-optimization starts to appear at a lower inference budget, and applied to TÜLU3-8B-SFT, our experiments show that QALIGN can consistently outperform direct preference optimization (DPO; Rafailov et al., 2024), which updates the model weights, as well as BoN, MV, and WMV on a diverse range of datasets in mathematical reasoning (**GSM8K** and **MATH500**; Hendrycks et al., 2021; Lightman et al., 2023), knowledge recall (**MMLU-Redux**; Gema et al., 2025), **TruthfulQA** (Lin et al., 2022) and instruction following (**IFEval**; Zhou et al., 2023b), as shown in Figure 1.[1]

Our contributions are:

- We propose QALIGN, a method for making local approximations of the optimal aligned distribution at test time.

- We show a $28.4\%$ increase in accuracy on GSM8K and $78.34\%$ on GSM-Symbolic relative to a single generation, using a task-specific RM, beating BoN, MV, and WMV.

- We show that QALIGN outperforms DPO when applied to TÜLU3 family of models, in general RMs trained on the TÜLU3 preference dataset, even when compared with the same compute budget at test-time via MV, achieving a $47.85\%$ increase in average accuracy relative to a single generation across a suite of benchmarks.

---

[1]Website with code and additional resources will be provided upon acceptance.

## 2 BACKGROUND: LANGUAGE MODEL ALIGNMENT

Finetuning a pretrained LM $p_{\text{LM}}(y \mid x)$ to align with preferences encoded by a reward function $r(y, x)$ can be cast as Bayesian inference (Korbak et al., 2022a). We seek $\pi(y \mid x)$, which is initially equal to a prior $p_{\text{LM}}(y \mid x)$, and updated to conform to evidence provided by the human preferences model. Let $\gamma = 1$ denote the event that response $y$ is maximally preferred for prompt $x$ (i.e., $y \succeq y'$ for all $y' \in \mathcal{Y}$). Assuming the reward function is bounded, we can model the likelihood of this event as $p(\gamma = 1 \mid y, x) = \exp\left((r(y, x) - \max_{y'} r(y', x))/\beta\right)$. Using Bayes' rule, the posterior takes the following form:

$$\pi(y \mid x) \triangleq p(y \mid \gamma = 1, x) = \tfrac{1}{Z_\beta(x)} p_{\text{LM}}(y \mid x) \exp\left(\tfrac{r(y,x)}{\beta}\right). \tag{1}$$

It is both intractable to compute the partition function $Z_\beta(x) = \sum_{y \in \mathcal{Y}} p_{\text{LM}}(y \mid x) \exp(r(y, x)/\beta)$, and to sample from the distribution in Eq. 1 exactly.

To approximate $\pi(y \mid x)$, many methods have been proposed (Stiennon et al., 2022; Fernandes et al., 2023; Kaufmann et al., 2023; Peters & Schaal, 2007; Peng et al., 2019; Korbak et al., 2022b;a; Go et al., 2023). Given a dataset of prompts $\mathcal{D}$, one common approach is to estimate a parameterized variational approximation $q_\theta(y \mid x)$:

$$\max_\theta \mathbb{E}_{x \sim \mathcal{D}} \left[ \mathbb{E}_{y \sim q_\theta(y|x)} \left[ r_\phi(y, x)/\beta \right] - D_{\text{KL}} \left( q_\theta(y \mid x) \parallel p_{\text{LM}}(y \mid x) \right) \right]. \tag{2}$$

Methods like PPO (Schulman et al., 2017) and others (Shao et al., 2024; Rafailov et al., 2024; Ahmadian et al., 2024) optimize this objective with low-variance gradient estimates, resulting in stable optimization. Importantly, $q_\theta(y \mid x)$ is a newly trained LM initialized with $p_{\text{LM}}(y \mid x)$.

## 3 TEST-TIME ALIGNMENT VIA MCMC

Casting language model alignment as posterior inference decouples our target goal from the procedure used to achieve it. While current approaches rely on variational inference to learn a single model $q_\theta(y \mid x)$ (Eq. 2), this strategy faces four limitations. First, it requires expensive model finetuning, which is costly with large models. Second, many models' weights are not openly shared, including those of state-of-the-art models like GPT-4 (OpenAI, 2024) and Gemini (Gemini, 2024). Third, amortizing the approximation across all prompts necessarily sacrifices the quality of approximation for any individual prompt $x$ to achieve good average performance. Fourth, the approach assumes a monolithic notion of human preferences encoded in $r(y, x)$, offering no flexibility to adapt to varying user preferences or contexts at inference time.

These limitations motivate a shift toward local posterior approximations, where we achieve a better approximation as we increase the compute budget at test time on a single prompt $x$. As we will describe in (§3.1), our key insight is that rather than estimating a single parametric approximation, we can use recent advancements in Markov chain Monte Carlo (MCMC) sampling with LMs (Faria et al., 2024) and obtain a sequence of samples $\mathcal{S} = \langle y^0, y^1, \ldots, y^T \rangle$ from $\pi(y \mid x)$ and use them for **optimal decision-making**.

A principled way to select a final response from the generated candidate set $\mathcal{S}$ from $\pi(y \mid x)$ is by selecting the most common element (mode) i.e., via majority voting (MV; Wang et al., 2023b). However, this only works for tasks with well-defined answers, such as mathematical reasoning or multiple-choice questions. For open-ended generation tasks, we apply a generalization of this approach through the minimum Bayes risk (MBR) principle (Kumar & Byrne, 2002; Eikema & Aziz, 2020; Farinhas et al., 2023; Bertsch et al., 2023). The MBR framework selects the output $\hat{y}$ that maximizes the expected task-specific utility $u(y, y')$:

$$\hat{y} = \arg\max_{y \in \mathcal{S}} \mathbb{E}_{y' \sim \pi(y'|x)} \left[ u(y, y') \right] \approx \arg\max_{y \in \mathcal{S}} \frac{1}{T+1} \sum_{t=0}^{T} u(y, y^t). \tag{3}$$

When the utility metric $u(y, y')$ is defined as an exact match over the final answer, MBR amounts to a MV strategy over $\mathcal{S}$. We employ ROUGE (Lin, 2004) as our utility metric for open-ended generation tasks, following Bertsch et al. (2023). While this results in $O(T^2)$ similarity computations, ROUGE is computationally lightweight.

In the following sections, we describe two practical approaches to obtain a final answer based on MBR to samples from $\pi(y \mid x)$. First, QALIGN (§3.1), our novel MCMC-based approach to sample from the aligned distribution $\pi(y \mid x)$ at test time, allowing us to apply MBR directly to these samples. Second, in §3.2, we present importance sampling, a classical alternative that circumvents direct sampling from the aligned distribution by reweighting samples from the base model $p_{\mathrm{LM}}(y \mid x)$ to approximate MBR as if they were drawn from $\pi(y \mid x)$. Finally, in §3.3, we describe best-of-$n$ sampling (BoN), a simple and widely used baseline approach for test-time alignment that does not explicitly optimize an MBR objective but instead selects a single high-reward sample from the base model.

## 3.1 MCMC FOR TEXT GENERATION

With the goal of generating a sequence of samples from $\pi_\beta(y \mid x)$, we will construct a Markov chain $(y^0, y^1, \dots, y^T)$ that has $\pi_\beta(y \mid x)$ as its equilibrium distribution. The chain starts from a hypothesis $y^0 \sim p_{\mathrm{LM}}(y \mid x)$. On the $t$th iteration, it draws a new hypothesis $y$ from a **proposal distribution** $q(y \mid y^t, x)$, and this hypothesis is accepted with an acceptance probability $\alpha_\beta(y, y^t) \in [0, 1]$. The **proposal** distribution $q(y \mid y^t, x)$ we use is the one proposed by QUEST (Faria et al., 2024). It is based on a LM that samples suffixes $p_{\mathrm{LM}}(y_{i:N} \mid y_{<i}, x)$ starting at uniformly sampled index $i$. This proposal can be written as

$$q(y \mid y^t, x, i) = p_{\mathrm{LM}}(y_{i:N} \mid y^t_{<i}, x) \times \mathbf{1}\{y_{1:i} = y^t_{1:i}\}, \tag{4}$$

Following the Metropolis-Hastings algorithm (MH; Hastings, 1970), the acceptance probability is

$$\alpha_\beta(y, y^t) = \min \left\{ 1, \ \pi_\beta(y \mid x) q(y^t \mid y, x) \ / \ \pi_\beta(y^t \mid x) q(y \mid y^t, x) \right\}. \tag{5}$$

If the candidate $y$ is accepted, the next state in the chain becomes $y^{t+1} = y$; if rejected, the chain stays at $y^{t+1} = y^t$. The process repeats for some number of steps $T$. In the end, it returns the set of accepted samples. Note that, while computing the likelihood $\pi_\beta(y \mid x)$ of a particular hypothesis $y$ under the target distribution is intractable (due to the partition function $Z_\beta(x)$), evaluating the acceptance criterion $\alpha_\beta(y, y^t)$ is easy, because it depends only on the likelihood ratio, in which the normalization constants cancel out:

$$\frac{\pi_\beta(y \mid x)}{\pi_\beta(y^t \mid x)} = \exp\left(\frac{r(y,x) - r(y^t,x)}{\beta}\right) \frac{p_{\mathrm{LM}}(y \mid x)}{p_{\mathrm{LM}}(y^t \mid x)}. \tag{6}$$

MH converges to the unique stationary distribution $\pi_\beta(y \mid x)$, regardless of the initial distribution, because the transition distribution of the Markov chain, $p(y^t \mid y^{t-1}, x)$ which results from generating a candidate from $q(y^t \mid y^{t-1}, x)$ followed by an accept/reject step, satisfies the *Markov chain ergodic theorem* (Neal, 2011).

Note that, under samples from the proposal from Eq. 4, the likelihood ratio from Eq. 6 is proportional to the inverse of the probability of returning:

$$\frac{q(y^t \mid y, x, i)}{q(y \mid y^t, x, i)} = \frac{p_{\mathrm{LM}}(y^t_{i:N} \mid y^t_{<i}, x)}{p_{\mathrm{LM}}(y_{i:N} \mid y^t_{<i}, x)}, \tag{7}$$

this allows simplifying the criterion as:

$$\alpha_\beta(y, y^t) = \min \left\{ 1, \exp\left(\frac{r(x,y) - r(x,y^t)}{\beta}\right) |y^t|/|y| \right\}. \tag{8}$$

The length ratio $|y^t|/|y|$ comes from the ratio between the uniform index distributions. Note that this means that we can sample from the aligned distribution in Eq. 1 without any access to logits or parameter weights.

In summary, we propose a simple and effective procedure for sampling from $\pi(y \mid x)$. We characterize the QALIGN sampling process as repeating the following for $T$ steps:

1. Given an instance $y^t$ with length $|y^t|$, sample an index $i$ uniformly.

2. Generate a completion $y_{i:N}$ from $p_{\mathrm{LM}}(y_{i:N} \mid y^t_{<i}, x)$.

3. Compute the probability of acceptance on the reward difference in Eq. 8. Sample a random boolean based on this probability. If we reject, $y^{t+1} = y^t$; if we accept, $y^{t+1} = y'$.

The QUEST proposal is simple and relies solely on the model's ability to generate text left-to-right, with the consequence that successive samples are only conditionally dependent up to the index $i$, which limits the speed at which we can explore (i.e., the effective sample size).

While more complex proposals could be built based on LMs prompted to self-refine (Zhou et al., 2023a; Madaan et al., 2023; Yao et al., 2023), recent work demonstrates that even strong LMs often fail to improve reasoning due to models' inability to gauge the correctness of their outputs and localize errors (Huang et al., 2024). Because of this, a growing body of research tries to teach LMs how to self-correct by creating data-augmentation recipes (Welleck et al., 2022; Havrilla et al., 2024) or new datasets (Wang et al., 2023a; Chen et al., 2024; Lee et al., 2024; Saunders et al., 2022; Schick et al., 2022). However, when the proposal is different from the base model, we lose the simplicity of the acceptance criterion in Eq. 8 and need to keep track of two sets of logits to calculate the acceptance criterion in Eq. 5.

As our approach only requires access to the ability to generate continuations given a prefix (not access to base model parameters or logits), it can also be used with closed LMs through an API. However, we limit our experiments to open-weight models. We provide in Appendix B an analysis of the computational cost associated with running QALIGN.

### 3.2 IMPORTANCE SAMPLING

A natural competing approach to QALIGN for approximating the intractable expectation in Eq. 3 is to use importance sampling (IS; Kahn, 1950). Rather than attempting to directly sample from the target distribution $\pi(y \mid x)$, IS generates samples from the base LM $p_{\text{LM}}(y \mid x)$ and reweights them to match the target distribution $\pi(y \mid x)$:

$$\mathbb{E}_{y \sim \pi(y|x)}[u(y, y')] = \mathbb{E}_{y \sim p_{\text{LM}}(y|x)}\left[\frac{\pi(y \mid x)}{p_{\text{LM}}(y \mid x)}u(y, y')\right] = \mathbb{E}_{y \sim p_{\text{LM}}(y|x)}\left[\frac{\exp(\frac{1}{\beta}r(y, x))}{Z_\beta(x)}u(y, y')\right].$$

The partition function $Z_\beta(x)$ can be reframed as an expectation over samples from the base model: $Z_\beta(x) = \sum_{y \in \mathcal{Y}} p_{\text{LM}}(y \mid x)\exp(r(y, x)/\beta) = \mathbb{E}_{y \sim p_{\text{LM}}(y|x)}[\exp(r(y, x)/\beta)]$. This motivates a self-normalizing importance sampling approach, where we approximate both the numerator and denominator using samples $\{y^{(i)}\}_{i=1}^K \sim p_{\text{LM}}(y \mid x)$. The self-normalized estimator is consistent, but introduces bias due to the correlation between the numerator and denominator (Owen, 2013). In the end, making this approximation boils down to the following steps:

1. Sample $y^{(0)}, \ldots, y^{(T)} \sim p_{\text{LM}}(y \mid x)$.

2. Evaluate the generations with the reward model $r(y, x)$ resulting in $r^{(0)}, \ldots, r^{(T)}$.

3. Obtain the importance weights $w^{(i)} = \exp(r^{(i)}/\beta) \left/ \sum_{j=0}^T \exp(r^{(j)}/\beta)\right.$.

4. For each hypothesis $y'$, compute $\mathbb{E}_{y \sim \pi(y|x)}[u(y, y')] \approx \sum_{i=0}^T w^{(i)}u(y, y^{(i)})$.

In the LM literature, this procedure, for the specific case of tasks with well-defined final answers (i.e., the utility metric $u(y, y')$ is defined as an exact match), the MBR output is the same as **weighted majority voting** (WMV; Li et al., 2023). Similar to QALIGN, WMV is guaranteed to converge to the optimal decision rule as computational resources increase. However, because it relies solely on independent samples drawn from the base LM, the generation process cannot be directly steered toward more promising regions of the output space. This limitation becomes particularly problematic when the base LM infrequently produces high-reward responses. In such cases, the approximation of the target expectation, and corresponding final decision, will be compromised.

### 3.3 BEST-OF-$n$ SAMPLING

Best-of-$n$ (BoN) sampling is a simple approach for aligning language model predictions at test time. BoN has emerged as a strong baseline in the literature, requiring no additional training, while still achieving compelling performance. Unlike QALIGN and IS, which explicitly optimize for a MBR objective, BoN uses a heuristic selection process.

Given a prompt $x$, we generate $n$ candidate responses $y_1, y_2, \ldots, y_n$ independently from the base LM $p_{\text{LM}}(y \mid x)$. Each response is evaluated using the RM $r(x, y)$, and the candidate with the highest reward is chosen as the final output:

$$y^*(n) = \arg\max_{y_i} r(x, y_i). \tag{9}$$

This procedure implicitly biases the output distribution toward higher-reward responses. It is easy to see that in the limit of $n$, we are sampling from the maximum reward distribution; however, BoN is remarkably performant even when $n$ is small (Fernandes et al., 2022; Gao et al., 2022; Llama, 2024).

With some strong assumptions, we can show that the probability density of the BoN distribution can be approximated by our target distribution $\pi(y \mid x)$ from Eq. 1 in the special case when $\beta$ takes the form:

$$\beta(n) = \frac{\sigma_d(x)}{\sqrt{2 \log n_d} - \frac{(\log \log n_d + \log(4\pi))}{2\sqrt{2 \log n_d}}}. \tag{10}$$

where $\sigma_d(x)$ is the standard deviation of the dominant component of the distribution of reward, under the assumption of Gaussian mixture, which we empirically observe (see Appendix C), when evaluated on samples from our base LM conditioned on $x$, and $n_d = w_d(x)n$, the dominant component's sample size, where $w_d(x)$ is the weight of the dominant component. A full derivation of this approximation is provided in Appendix A.

This rough equivalence, not noted in past literature to our knowledge, provides a new insight into a key distinction between BoN and the other methods we have discussed, including QALIGN. In QALIGN, we explicitly define a target distribution $\pi(y \mid x)$, and increasing the test-time budget $n$ refines our approximation of this fixed objective. In contrast, BoN does not converge to a predefined distribution; instead, increasing $n$ progressively shifts the selection process toward responses with higher rewards and deviating further from $p_{\text{LM}}(y \mid x)$. Insofar as the reward model is well-aligned with human preferences, this should make BoN highly effective. However, in realistic settings where the reward is imperfect, increasing $n$ can lead to over-optimization, i.e., selecting responses that score well under the reward model but degrade in actual quality. As a result, BoN can suffer from diminishing returns or even performance degradation as test-time compute increases, which we observe in our experiments (§4.1–4.2) and previous work (Gao et al., 2022).

## 4 EXPERIMENTS

Our experiments are centered around two main questions. First, **task-specific tuning:** given a powerful task-specific RM trained on preference data derived from ground-truth evaluations of base model outputs, can QALIGN outperform BoN and WMV (§4.1)? Second, **general alignment:** when applied as an alternative to DPO, where the preference dataset encompasses multiple tasks aimed at improving the instruction-following capabilities of chat models, can QALIGN outperform state-of-the-art alignment methods (§4.2)?

### 4.1 TASK-SPECIFIC TUNING

We evaluate QALIGN for task-specific tuning using LLAMA-3.1-8B-INSTRUCT (Llama, 2024) as the base model and a custom Bradley & Terry (1952) RM trained on our preference dataset for mathematical reasoning (training details in Appendix F). This RM was initialized with LLAMA-3.1-8B-INSTRUCT and trained on 64 model-generated response pairs per prompt. We selected pairs based on ground truth answers.[2]

**Baselines** We compare against BoN, and WMV sampling applied to LLAMA-3.1-8B-INSTRUCT using our trained RM and MV from samples from LLAMA-3.1-8B-INSTRUCT. For each of the methods we generate 1024 solutions per problem.

**Datasets** We evaluate on **GSM8K** (Cobbe et al., 2021), a dataset of grade school math problems, and **GSM-Symbolic** (Mirzadeh et al., 2024), a new benchmark designed to assess out-of-distribution generalization in mathematical reasoning. As demonstrated by Mirzadeh et al. (2024), GSM-Symbolic

---

[2]Data and RM will be provided upon publication.

exposes substantial performance degradation across state-of-the-art LLMs exhibiting drops of up to $65\%$ compared to their GSM8K scores when faced with simple variations like changed numerical values and extra redundant clauses.

**Sampling configurations**   We sampled both datasets with temperature $1.0$ and $\beta = 1$. Since the acceptance rate is a function of $\sigma_r(x)/\beta$ (where $\sigma_r(x)$ is the distribution of rewards under the base model), we selected this value to achieve approximately $50\%$ acceptance rate based on tuning with $128$ samples from the GSM8K training data. This relatively high acceptance rate ensures the chain mixes well and avoids getting stuck in a single mode.

**Results**   The results of the task-specific tuning experiments are summarized in Figure 2. QALIGN shows progressively lower error rates across both problems as we spend more computational resources. In contrast, on GSM8K, MV shows initial improvement but quickly saturates, while BoN displays improvement with additional compute, temporarily outperforming QALIGN, but eventually reaches an inflection point, after which error rates begin to increase. This behavior aligns with the observations from Gao et al. (2022) and theoretical results from §3.3. WMV performs well on GSM8K but becomes relatively worse after a budget of approximately $2^8 \times 10^{12.51}$ FLOPs.

While all methods show some performance drop on GSM-Symbolic compared to GSM8K, the magnitude varies significantly. BoN and WMV, which rely on the RM, struggle more with the distribution shift, likely because the RM is fit to particular GSM8K-specific patterns. Relative to these two, MV shows greater robustness to these changes. This is primarily because under the distribution shift the RM becomes less reliable, and since MV does not use it, MV's performance degrades less. However, even under these circumstances, QALIGN is able to leverage the RM and outperform MV on this dataset with enough compute. This suggests that QALIGN is robust to imperfections in the RM and able to extract a useful signal even when the RM's reliability is compromised.

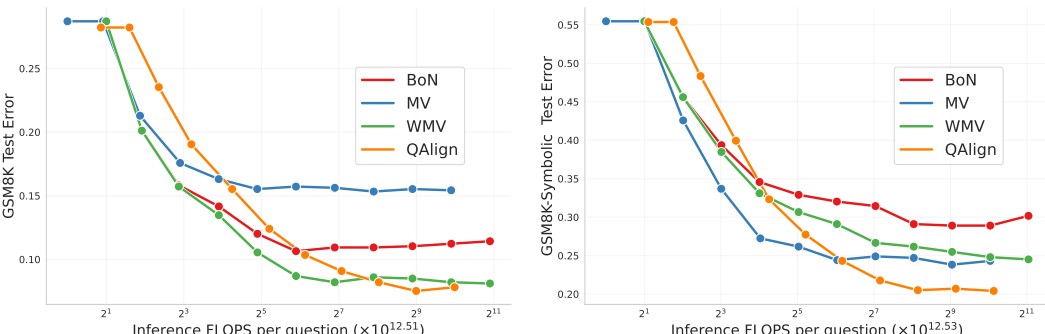

Figure 2: Average accuracy vs. floating point operations (FLOPS) in log scale. We compare ● QALIGN with LLAMA-3.1-8B-INSTRUCT against three baselines also applied to LLAMA-3.1-8B-INSTRUCT: ● **best-of-$n$ (BoN)**, ● **majority vote (MV)**, and ● **weighted MV (WMV)**. *Left*: Error rate (lower is better) on GSM8K test dataset. *Right*: Error rate on GSM-Symbolic test dataset. All experiments use temperature 1.0 with reasoning included in model outputs.

## 4.2   GENERAL ALIGNMENT

We evaluate QALIGN for more general alignment using TÜLU3-8B-SFT as the base model and TÜLU3-8B-RM as the RM. The TÜLU3 model family (Lambert et al., 2024) was selected for being fully open source, providing instruction-tuned, and aligned versions, along with their corresponding training code and datasets. This enables direct comparison of QALIGN on top of the instruction-tuned model to the aligned model, sharing the initial starting point. Additionally, they provide an RM trained on the data used to train the DPO model, which we use in our experiments.

**Baselines**   We compare against (1) MV applied to TÜLU3-8B-DPO, (2) BoN sampling applied to TÜLU3-8B-SFT using TÜLU3-8B-RM, (3) WMV applied to TÜLU3-8B-SFT using TÜLU3-8B-RM, and (4) MV applied to TÜLU3-8B-SFT. Note that TÜLU3-8B-DPO model is the result of doing preference finetuning on the TÜLU3-8B-SFT with 271k preference pairs (approximately $1.75 \times 10^{19}$

| Method | MATH500 | GSM8K | TQA | MMLU-Redux | IFEval | Avg |
|---|---|---|---|---|---|---|
| | (0 shot, CoT) | (0 shot, CoT) | (MC2, 0 shot, CoT) | (0 shot, CoT) | (prompt loose) | |
| **TÜLU3-8B-SFT** | | | | | | |
| BoN | 31.6 | 74.3 | 45.3 | 57.1 | 59.3 | 53.5 |
| MV | 49.4 | 86.3 | 43.7 | 59.4 | 72.6 | 62.3 |
| WMV | 53.0 | 85.2 | 46.1 | 62.2 | 49.5 | 59.2 |
| QALIGN | **60.4** | **88.2** | **48.5** | 62.2 | **77.6** | **67.4** |
| **TÜLU3-8B-DPO** | | | | | | |
| MV | 55.8 | 87.9 | 45.4 | **62.7** | 76.2 | 65.6 |

Table 1: **Overview of the results on general alignment.** Results show accuracy percentages with fixed 1024 sampled solutions across different methods and models. Our proposed QALIGN outperforms other approaches on most benchmarks, achieving the highest scores on all datasets when applied to TÜLU3-8B-SFT, and attaining an average performance better than TÜLU3-8B-DPO. The highlighted cells indicate best result per benchmark.

FLOPs). Comparing against DPO with MV additionally allows for an inference FLOPs-adjusted comparison between a model explicitly optimized for alignment and our QALIGN approach. This allows us to test whether QALIGN is not only a better test-time alignment approach, but a better overall alignment approach when significant compute is available at test-time.

**Datasets** We evaluate again on **GSM8K**; we add **MATH500**, a verified subset of the MATH test dataset that contains advanced mathematical problems across algebra, geometry, and calculus (Hendrycks et al., 2021), **MMLU-Redux** (Gema et al., 2025), a refined subset of commonsense and academic knowledge questions, **TruthfulQA** (Lin et al., 2022), which contains misleading questions designed to elicit truthful responses, and **IFEval** (Zhou et al., 2023b), which measures adherence to complex multi-step instructions. Among these datasets, IFEval is an open-ended generation and, therefore, requires the use of MBR with ROUGE-1 instead of MV and WMV. For simplicity, we still refer to the method as MV and WMV to denote MBR from base model samples and the weighted version with importance sampling, respectively, in our comparisons.

**Sampling Configurations** For all datasets, we sampled the model with a temperature of 1.0 and prompted it to provide reasoning. Complete prompts are available in Appendix D. As with the mathematical reasoning experiments, we tuned the $\beta$ parameter (in this case, $\beta = 0.5$), for TÜLU3-8B-RM, using 128 samples from the GSM8K training data to achieve an average acceptance rate of 50%, and using the same $\beta$ for all datasets.

**Results** The results of the general alignment experiments are summarized in Table 1. Figure 1 plots the average error rate across all of the datasets as a function of the floating point operations (FLOPS), and Appendix E contains all of the error plots for each individual problem. Similar to our task-specific findings, QALIGN, MV, and WMV consistently reduce error rates across all five general alignment datasets as computation increases, with MV exhibiting early saturation. However, similar to the results from GSM-Symbolic, MV outperforms both WMV and BoN. MV applied to the DPO model begins with lower error rates but saturates earlier, ultimately reaching error rates that are worse than QALIGN's final results. Furthermore, we observe that BoN's inflection point occurs at a lower computational budget than observed in task-specific experiments. This suggests that for real-world alignment problems, BoN is not an effective test-time approach, while QALIGN maintains its improvement trajectory even with general RMs.

## 5 RELATED WORK

**MCMC for Text Generation.** Our work builds on QUEST (Faria et al., 2024), which uses MCMC to sample diverse translations. While QUEST successfully generates multiple translations, its effectiveness as a selection mechanism for a single high-quality response remained unexplored until now. Earlier works have applied MCMC approaches to both autoregressive and masked language

models (MLMs). For instance, Miao et al. (2018) and Zhang et al. (2020) use MH with a proposal distribution that makes token-level modifications for constrained generation tasks.

In the case of MLMs, previous works have explored various forms of Gibbs sampling (Berglund et al., 2015; Su et al., 2018; Wang & Cho, 2019; Yamakoshi et al., 2022). However, as Goyal et al. (2022) show, the conditional distributions from MLMs result in invalid Gibbs samplers. In response, they propose an MH correction on the masked conditionals, resulting in higher quality generations. Building on this, Mireshghallah et al. (2022) and Forristal et al. (2023) apply MLMs for controlled text generation. Furthermore, several works have adapted Hamiltonian MCMC algorithms originally designed for high-dimensional continuous distributions (Duane et al., 1987; Neal, 2011) to discrete text generation (Kumar et al., 2022; Qin et al., 2022; Amini et al., 2023; Du et al., 2023).

**Test-Time Scaling.**    Test-time scaling methods have emerged as an important approach for improving LM performance without additional training. While our work focuses on local approximations to the optimal aligned distribution from Eq. 1, a parallel line of research explores heuristic search strategies. Best-of-$n$ (BoN; Gao et al., 2022; Stiennon et al., 2020), majority voting (MV; Wang et al., 2023b), weighted majority voting (WMV; Li et al., 2023) are examples of such approaches. As we outline in§ 3.2–3.3, BoN and WMV can be interpreted as doing test-time alignment.

The RMs we consider in this work are outcome-based RMs, they only work on full generations. Several recent methods use process-based RMs (PRMs) that evaluate partial generations (Lightman et al., 2023; Wang et al., 2024; Uesato et al., 2022). Many techniques have been developed to take advantage of PRMs through guided beam search (Xie et al., 2023), Monte-Carlo tree search (MCTS; Liu et al., 2024; Zhang et al., 2023), and REBASE (Wu et al., 2024). However, as outlined in DeepSeek-AI (2025), PRMs face significant practical limitations.

Furthermore, recent theoretical work has analyzed test-time scaling methods, particularly BoN sampling. Farinhas et al. (2025); Schaeffer et al. (2025) derive BoN scaling laws relating the number of generated samples to performance gains. Gui et al. (2024) prove that BoN is optimal for the trade-off between win-rate against the base model and KL divergence from the base model. Beirami et al. (2025) provide win-rate guarantees and derive a closed-form probability mass function for the BoN policy, along with a new KL estimator. Yang et al. (2024) show that BoN is asymptotically equivalent to the optimal policy for reward maximization under KL constraints, assuming a memoryless LM and linear reward function. Our work extends this research direction by establishing a novel approximation connecting the BoN distribution with $n$ samples to the optimal aligned distribution with parameter $\beta$ (§ 3.3).

## 6 Conclusion and Future work

We introduced QALIGN, a test-time alignment method that can sample from the optimal aligned distribution without any model retraining (requiring only a reward model). Our empirical results consistently demonstrate that QALIGN outperforms both search-based approaches that attempt to maximize imperfect reward models (like BoN) and principled alternatives that rely on independent samples from the base LM (such as WMV and MV). Additionally, QALIGN outperforms DPO-tuned models even when they are allowed to match QALIGN's compute budget at test-time.

By enabling strong alignment results without model retraining, QALIGN opens new possibilities for deploying and improving language models in resource-constrained environments and enabling the use of private RMs with closed-source LMs.

Looking ahead, we believe this work sets the stage for further exploration of test-time alignment methods. While we have demonstrated success across several benchmarks, several important directions remain unexplored. These include the development of more sophisticated proposal distributions and using MCMC as a tool to understand the meta-generation strategies that emerge in RL-trained reasoning models.

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

## A    BEST-OF-$n$ SAMPLING AS TEST-TIME ALIGNMENT

Best-of-$n$ (BoN) sampling is a widely used technique for improving model performance by selecting the highest scoring sample among $n$ candidates. In this section, we show that BoN sampling can be interpreted as an implicit form of test-time alignment that approximates the **mode** of the target distribution $\pi(y \mid x)$ in Eq. 1. In §A.1, we derive the probability density function (pdf) for BoN sampling. Next, in §A.2, we use extreme value theory to obtain a tractable approximation of rewards of this distribution, and in §A.3 derive an expression for the distribution of rewards of the target density $\pi(y \mid x)$. Finally, in §A.4, employing a Laplace approximation of the reward distribution, we establish an explicit relationship between $n$ and the effective parameter $\beta$ of the aligned distribution $\pi(y \mid x)$.

### A.1    DISTRIBUTION OF MAXIMUM REWARD

In BoN sampling, given a prompt $x$, we generate $n$ candidate responses $y_1, y_2, \ldots, y_n$ independently from the base LM $p_{\text{LM}}(y \mid x)$. Each response is evaluated using the RM $r(y, x)$, and the candidate with the highest reward is chosen as the final output:

$$y^*(n) = \arg \max_{y_i} r(x, y_i). \tag{11}$$

Let us assume that the reward function $r(x, y)$ is (locally) injective, that is, each $y$ maps to a unique reward value. Then, if we denote by $p(r \mid x)$ the probability density (pdf) of reward values for a particular $x$ under the base LM $p_{\text{LM}}(y \mid x)$ a change of variables yields:

$$p(r \mid x) = p_{\text{LM}}\Big(y = r^{-1}(r, x) \mid x\Big) \Big| J_{r^{-1}}(r) \Big|, \tag{12}$$

where $r^{-1}$ is the inverse mapping and $|J_{r^{-1}}(r)|$ is its Jacobian determinant.

Given $n$ independent samples, the cumulative distribution function (cdf) $F_R(r|x)$ we can write the cdf for the maximum reward $r_{\max}$ as

$$F_{\text{bon}}(r_{\max}^{(n)}|x) = F_R(r_{\max}^{(n)} \mid x)^n, \tag{13}$$

and the respective pdf as

$$p_{\text{bon}}(r_{\max}^{(n)}|x) = n\, F_R(r_{\max}^{(n)} \mid x)^{n-1}\, p(r = r_{\max}^{(n)} \mid x). \tag{14}$$

Since each candidate $y$ corresponds to a reward $r(x, y)$, the probability density over the best-of-$n$ response can be written as:

$$p(y^*(n)|x) = n\, F_R\big(r(x, y^*(n)) \mid x\big)^{n-1}\, p_{\text{LM}}(y^*(n)|x). \tag{15}$$

Eq. 15 shows how the BoN procedure emphasizes the upper tail of the reward distribution. The term $nF_R(r \mid x)^{n-1}$ becomes increasingly significant as $n$ grows, effectively pushing the probability mass towards reward values for which $F_R(r \mid x) \approx 1$.

### A.2    EXTREME VALUE THEORY

Recall that our goal was to establish an explicit relationship between the number of samples we maximize over $n$ and the parameter $\beta$ of the aligned distribution in Eq. 1. One way to establish this relationship is through the reward distribution. However, even assuming $p(r \mid x)$ is normal, the mode of the distribution in Eq. 14 lacks a closed-form expression.

Following classic results from extreme value theory (Fisher & Tippett, 1928; David & Gumbel, 1960), we know that the distribution of the maximum (appropriately rescaled) converges to a Gumbel distribution. For the specific case where we assume that for a given prompt $x$ the reward function values are distributed as a mixture of two Normal distributions with means $\mu_1(x)$ and $\mu_2(x)$, variances $\sigma_1^2(x)$ and $\sigma_2^2(x)$, and mixture weights $w_1(x)$ and $w_2(x)$ (where $w_1(x) + w_2(x) = 1$). Appendix C provides empirical evidence that this assumption is reasonable. The tail behavior is dominated by the Gaussian component with the heavier tail, typically the one with larger variance (or larger mean if variances are equal).

$$1 - F_{r_{\max}^{(n)}}(r) \approx w_d(x)\left(1 - \Phi\left(\frac{r - \mu_d(x)}{\sigma_d(x)}\right)\right), \tag{16}$$

where $\Phi$ is the standard normal CDF, $d$ the index of the component with the largest variance, i.e., $d = \arg\max_i \sigma_i^2(x)$, or if variances are equal, $d = \arg\max_i \mu_i(x)$ is the index of the component with the largest mean.

The work of Hall (1979) provides asymptotic expressions for location and scale parameters. In particular, one finds that the maximum reward is approximately:

$$r_{\max}^{(n)} \approx a_n \equiv \mu_d(x) + \sigma_d(x)\left(\sqrt{2\log n_d} - \frac{(\log\log n_d + \log(4\pi))}{2\sqrt{2\log n_d}}\right), \tag{17}$$

and fluctuations about this location are on the order of $b_n \approx \frac{\sigma_d(x)}{\sqrt{2\log n_d}}$, and where $n_d = n w_d(x)$ denotes the effective sample count from the dominant normal component. So in the limit of $n \to \infty$ the distribution of maximum reward can be expressed as:

$$p(r_{\max}^{(n)}|x) \approx \text{Gumbel}(a_n, b_n). \tag{18}$$

Figure 3 compares this Gumbel approximation to the empirical distribution of maximum rewards, showing a close fit for $n \geq 32$.

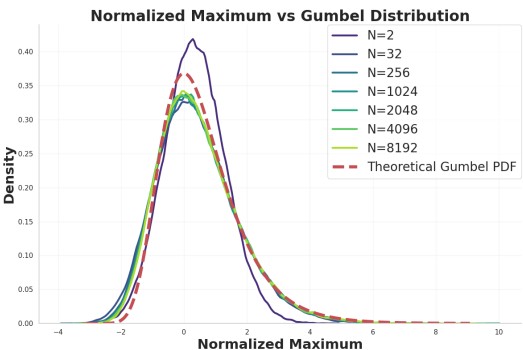

Figure 3: Distribution of the normalized maximum reward $(r_{\max}^{(n)} - a_n)/b_n$ for varying $n$, overlaid with the standard Gumbel distribution. The empirical distribution is estimated using 10,000 trials, each consisting of $n$ random samples drawn from a Normal distribution. The fit between the empirical distribution and the Normal distribution improves as $n$ increases, showing good agreement for $n \geq 32$.

### A.3 DISTRIBUTION OF REWARDS OF $\pi(y \mid x)$

Given the aligned distribution $\pi(y \mid x)$ from Eq. 1, we want an expression for the distribution of rewards under samples from $\pi$. Following the empirical observations from Appendix C, let us assume that for a given prompt $x$, the reward values under the base language model $p_{\text{LM}}(y \mid x)$ follow a mixture of two Normal distributions with means $\mu_1(x)$ and $\mu_2(x)$, variances $\sigma_1^2(x)$ and $\sigma_2^2(x)$, and mixture weights $w_1(x)$ and $w_2(x)$ (where $w_1(x) + w_2(x) = 1$), i.e.:

$$p(r \mid x) = w_1(x)\frac{1}{\sqrt{2\pi\sigma_1^2(x)}}\exp\left(-\frac{(r-\mu_1(x))^2}{2\sigma_1^2(x)}\right) + w_2(x)\frac{1}{\sqrt{2\pi\sigma_2^2(x)}}\exp\left(-\frac{(r-\mu_2(x))^2}{2\sigma_2^2(x)}\right).$$

Under $\pi(y \mid x)$, the distribution of rewards can be derived through a change of variables. The probability density of rewards under the policy, denoted as $\pi(r \mid x)$, is proportional to:

$$\pi(r \mid x) \propto \exp\left(\frac{r}{\beta}\right) p(r \mid x). \tag{19}$$

Substituting the mixture form of $p(r \mid x)$ and combining terms in the exponent and through completion of the square in the exponents for each component, we can show that this is equivalent to:

$$\pi(r \mid x) \propto w_1(x)C_1\frac{1}{\sqrt{2\pi\sigma_1^2(x)}}\exp\left(-\frac{1}{2\sigma_1^2(x)}\left(r - \mu_1(x) - \frac{\sigma_1^2(x)}{\beta}\right)^2\right) \tag{20}$$

$$+ w_2(x)C_2\frac{1}{\sqrt{2\pi\sigma_2^2(x)}}\exp\left(-\frac{1}{2\sigma_2^2(x)}\left(r - \mu_2(x) - \frac{\sigma_2^2(x)}{\beta}\right)^2\right), \tag{21}$$

where $C_1 = \exp\left(\frac{\mu_1(x)}{\beta} + \frac{\sigma_1^2(x)}{2\beta^2}\right)$ and $C_2 = \exp\left(\frac{\mu_2(x)}{\beta} + \frac{\sigma_2^2(x)}{2\beta^2}\right)$.

Therefore, the aligned distribution of rewards remains a mixture of Normals with adjusted mixture weights and means, and preserved variances:

$$\pi(r \mid x) = w_{\pi,1}\mathcal{N}(r; \mu_{\pi,1}(x,\beta), \sigma_1^2(x)) + w_{\pi,2}\mathcal{N}(r; \mu_{\pi,2}(x,\beta), \sigma_2^2(x)), \tag{22}$$

where $\mu_{\pi,i}(x,\beta) = \mu_i(x) + \sigma_i^2(x)/\beta$ for $i \in \{1,2\}$, and the adjusted mixture weights are given by:

$$w_{\pi,1} = \frac{w_1(x)C_1}{w_1(x)C_1 + w_2(x)C_2}, \quad w_{\pi,2} = \frac{w_2(x)C_2}{w_1(x)C_1 + w_2(x)C_2}. \tag{23}$$

This result shows that $\beta$ shifts the mean of each component by a factor proportional to that component's variance and inversely proportional to the parameter $\beta$, while preserving the variances of the original reward distributions. Additionally, as $\beta$ approaches zero, the mixture weight of the component with the larger variance (or larger mean if variances are equal) approaches 1, **causing the aligned distribution to collapse to a single Gaussian with an increasingly high mean**.

For this reason, we write the reward target density expression as approximately only the dominant Gaussian:

$$\pi(r \mid x) \approx \mathcal{N}(r; \mu_{\pi,d}(x,\beta), \sigma_d^2(x)), \tag{24}$$

where $d = \arg\max_i \sigma_i^2(x)$ is the index of the component with the largest variance, or if variances are equal, $d = \arg\max_i \mu_i(x)$ is the index of the component with the largest mean.

### A.4 RELATING $n$ AND $\beta$ VIA MODE MATCHING

The Gumbel approximation has a mode that can be explicitly expressed as its location parameter $a_n$, and the reward distribution of the aligned distribution in Eq. 24 as $\mu_{\pi,d}(x,\beta)$.

Following the results from Appendix A.3, the distribution of rewards under the target density $\pi(r|x)$ is also normal, but with mean $\mu_{\pi,d}(x,\beta) = \mu_d(x) + \sigma_d(x)^2/\beta$.

By matching this mode with the location parameter $a_n$ of the Gumbel distribution for BoN sampling i.e.:

$$\mu_d(x) + \frac{\sigma_d(x)^2}{\beta} = \mu_d(x) + \sigma_d(x)\left(\sqrt{2\log n_d} - \frac{(\log\log n_d + \log(4\pi))}{2\sqrt{2\log n_d}}\right), \tag{25}$$

we obtain:

$$\beta^* = \frac{\sigma_d(x)}{\sqrt{2\log n_d} - \frac{(\log\log n_d + \log(4\pi))}{2\sqrt{2\log n_d}}}. \tag{26}$$

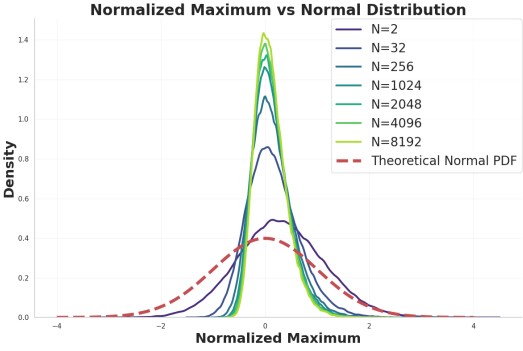

Figure 4: Distribution of the normalized maximum reward $(r_{\max}^{(n)} - \mu_{\pi,d}(x,\beta^*))/\sigma_d(x)$ for varying $n$, overlaid with the standard Normal distribution. The empirical distribution is estimated using 10,000 trials, each consisting of $n$ random samples drawn from a Normal distribution. While the mode of our approximation matches, the approximation does not capture the variance of the empirical distribution.

Figure 4 compares the empirical reward distribution to the target distribution $\pi(r|x)$ when $\beta = \beta^*$. The approximation closely matches the mode of the empirical distribution of maximum rewards, though it struggles to capture the variance accurately. The variance of the BoN reward distribution decreases as a function $n$, while the one from $\pi(y \mid x)$ stays constant.

## B  COMPUTATIONAL COST OF QALIGN

As outlined in Faria et al. (2024), when using the suffix proposal distribution from QUEST, each step samples, on average, an index at the midpoint of the sentence from the uniform distribution. Assuming a fixed sentence length $N$ for simplicity, this requires generating only $\frac{N}{2}$ new tokens on average per step. With typical transformer-based models, this allows us to reuse the majority of the key-value cache from previous iterations in both the base model and reward model. The reuse extends even beyond half of the computation due to the static prompt. However, unlike when generating independent samples, the computation for each prompt is sequential. We need to first generate the sentence $y^t$ before generating the sentence $y^{t+1}$. This means that regardless of the compute capability, compared with sampling independent samples, there is always an inherent latency overhead.

In summary, for a chain of $T$ steps, QALIGN is expected to generate $\frac{(T+1)N}{2}$ tokens in total: $N$ tokens for the initial hypothesis, plus an average of $\frac{N}{2}$ tokens for each of the remaining $T-1$ steps. In comparison, generating $T$ independent samples requires decoding $T \times N$ tokens. Therefore, for an equal number of samples, QUEST requires approximately half as many tokens from generation alone than sampling independently, translating to roughly **half the FLOPs**.

## C  EMPIRICAL OBSERVATIONS ON DISTRIBUTION OF REWARDS

When analyzing reward model predictions across independently generated responses from the base model for individual prompts, we consistently observed a bimodal distribution forming a two-component Normal mixture with distinct means and variances (Figure 5). This pattern appeared across the reward models used.

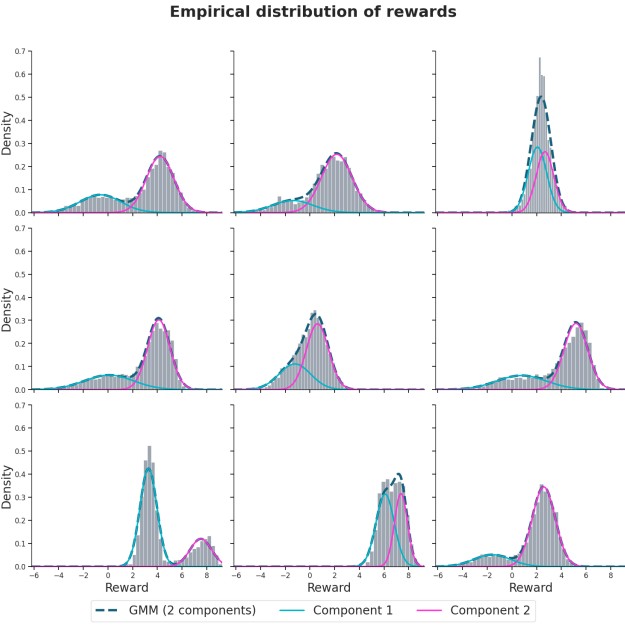

Figure 5: Histogram of rewards assigned by TÜLU3-8B-RM to $1,024$ responses generated by TÜLU3-8B-SFT for 9 randomly sampled prompts from GSM8K. For each prompt, we fit a two-component Gaussian mixture model to characterize the reward distribution.

We suspect this bimodal structure directly relates to the Bradley-Terry training objective, which naturally tries to separate the responses into two clusters (preferred vs. non-preferred). The clusters appear to have a normal distribution most likely because of the central limit theorem, i.e., the predictions result from a neural net that is the sum of billions of random variables.

# D   PROMPTS USED FOR EVALUATION

This appendix documents the prompt templates used across different datasets in our experiments. The prompt for GSM8K is only used in the general alignment experiments. The placeholders (text within <{...}>) are dynamically replaced with specific content from each dataset during the experiments.

## D.1   GSM8K DATASET

```
Solve the following grade school math problem step-by-step:
<{question}>
```

## D.2   MATH-500 DATASET

```
Solve the following math problem step-by-step:  <{question}>
Present the answer in LaTex format:  \boxed{Your answer}
```

## D.3   MULTIPLE CHOICE DATASETS (TQA, MMLU)

```
Choose the correct answer to the following multiple-choice
question about <{subject}>.
Question:  <{question}>
A). <{choice_A}>
B). <{choice_B}>
C). <{choice_C}>
D). <{choice_D}>
Provide your reasoning about the answer and finish your
answer with the letter corresponding to the correct option
(e.g., A, B, C, or D).
```

# E  FULL GENERAL ALIGNMENT PLOTS

Figure 6 plots the average error rate across all of the datasets as a function of the floating point operations (FLOPS).

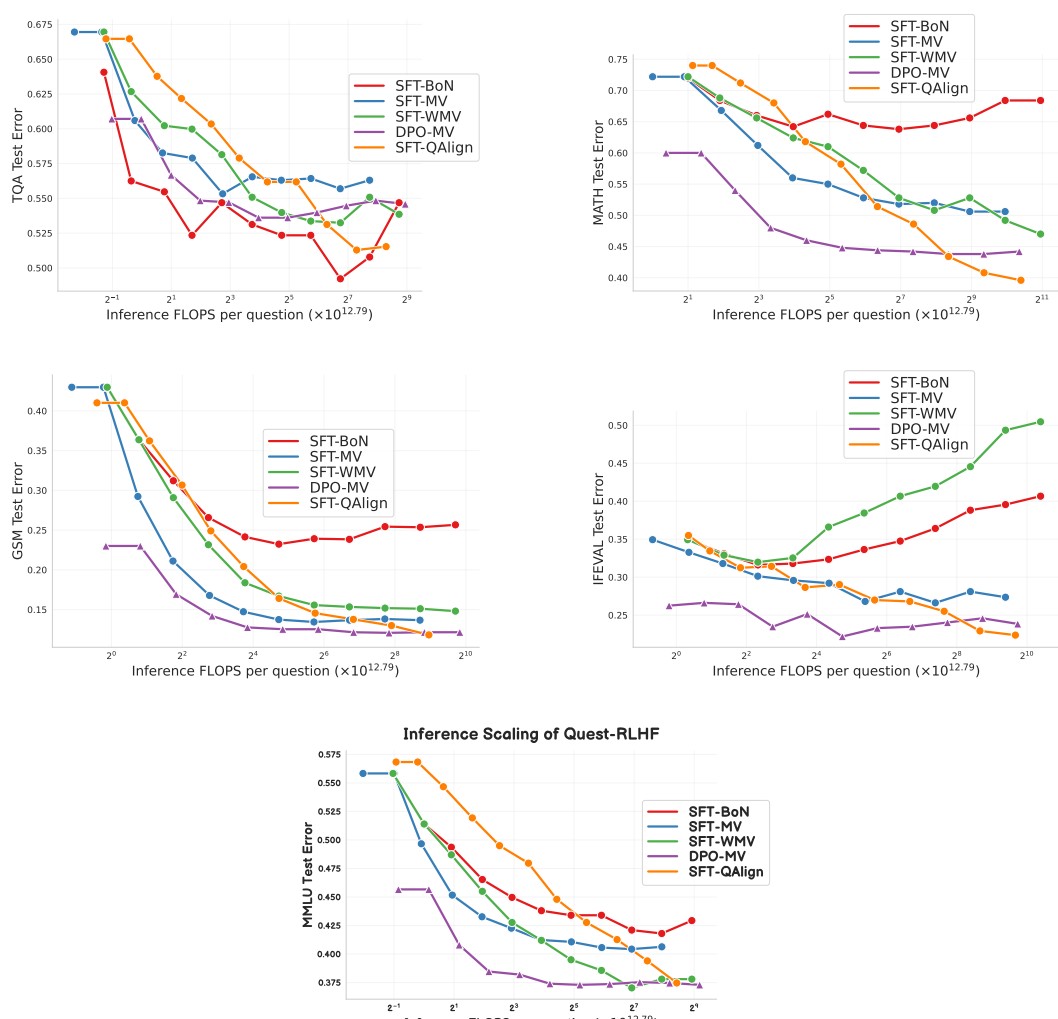

Figure 6: Error rate across multiple evaluation datasets (GSM8K, MATH500, MMLU-Redux, TruthfulQA, and IFEval) as a function of the floating point operations (FLOPS) in log scale. We compare ●QALIGN with TÜLU3-8B-SFT against four baselines: ▲ majority vote (MV) TÜLU3-8B-DPO, and applied to TÜLU3-8B-SFT the methods ● best-of-$n$ (BoN), ● MV, and ● weighted MV (WMV) . All experiments use temperature 1.0 with reasoning included in model outputs. Note that TÜLU3-8B-DPO model is the result of doing preference finetuning on the TÜLU3-8B-SFT with 271k preference pairs. The costs associated with this process are not accounted for in this plot.[GF: change plot ]

# F  REWARD MODEL TRAINING CONFIGURATION

We trained a Bradley & Terry (1952) reward model using LLAMA-3.1-8B-INSTRUCT (Llama, 2024) as the base model on a preference dataset for mathematical reasoning. The dataset contained 64 model-generated responses per prompt from GSM8K (Cobbe et al., 2021) (on-policy data from LLAMA-3.1-8B-INSTRUCT), from which we constructed one preference pair per prompt by selecting one ground truth correct answer as preferred and one incorrect answer as dispreferred. Training hyperparameters are detailed in Table 2.

Table 2: Training Hyperparameters for the RM for the task-specific experiments.

| Parameter | Value |
|---|---|
| **Model Configuration** | |
| Fine-tuning Type | Full |
| Chat Template | llama3 |
| **Dataset** | |
| Validation Split | 5% |
| **Optimization** | |
| Learning Rate | $1 \times 10^{-5}$ |
| Total Batch Size | 128 |
| Training Epochs | 1.0 |
| LR Scheduler | Linear Decay |
| Warmup Ratio | 0.03 |
| Weight Decay | 0.0 |
| Precision | bfloat16 |
| **Infrastructure** | |
| Optimization Framework | DeepSpeed ZeRO Stage 3 |
| GPUS | 8 L40S |

