# OpenReview forum: "Sample, Don't Search: Rethinking Test-Time Alignment for Language Models"
_ICLR.cc/2026/Conference — Submitted to ICLR 2026_

### Official Review · Reviewer_J4U9 · 2025-10-24

**Soundness:** 2
**Presentation:** 2
**Contribution:** 2
**Rating:** 4
**Confidence:** 4

**Summary:**

This paper introduces QALIGN, a test-time alignment method for language models that leverages Markov Chain Monte Carlo (MCMC) sampling to approximate the “optimal aligned distribution” without finetuning model parameters. The method adapts the QUEST algorithm to use a reward model (RM) for preference-based acceptance–rejection sampling, allowing aligned text generation with closed or open-weight models. Experiments on mathematical reasoning (GSM8K, GSM-Symbolic) and general instruction-following (MMLU-Redux, TruthfulQA, IFEval, MATH500) demonstrate consistent gains over best-of-n (BoN), majority voting (MV), weighted MV (WMV), and even Direct Preference Optimization (DPO) under equivalent compute budgets.

**Strengths:**

- The paper identifies an important limitation in current test-time search methods, over-optimization of imperfect RMs when scaling compute, and proposes a principled sampling-based alternative.

- Results are shown across both task-specific and general alignment settings, demonstrating robustness to RM imperfections and outperforming multiple baselines.

**Weaknesses:**

While the paper is clearly written and technically sound, several aspects could be clarified or strengthened to better support its claims:

- The method can be viewed as a length-normalized, RM-guided extension of BoN using MCMC proposals from QUEST. While the MH-based acceptance criterion is well-formulated, it is not entirely clear that this constitutes a fundamentally new paradigm rather than a reparameterization of BoN/MBR with additional tuning. In addition, since QALIGN generates samples sequentially, it forgoes the parallel efficiency of BoN, which raises questions about its overall computational practicality.

- The evaluation focuses on a single policy model (LLAMA-3.1-8B-Instruct) and one reward model (TÜLU3-8B-RM), leaving it uncertain whether the observed improvements would generalize across different architectures or RM designs.

- The main comparison in Table 1 uses N = 1024 samples for both BoN and QALIGN, which is considerably higher than typical inference budgets (e.g., N = 16–32). Results under more practical compute settings would strengthen the study. Furthermore, the FLOPs-adjusted comparison with DPO is not fully equitable, as training FLOPs are a one-time cost whereas inference FLOPs recur for each query.

**Questions:**

- A natural baseline, sampling multiple random indices and running BoN over them, is not tested, making it unclear whether MCMC brings tangible benefit.

- Can the authors offer more details regarding FLOPs computation for inference-time and training-time methods and how do they align?

- The related work section is relatively comprehensive but misses several works on inference-time alignment, such as:

1. Fast Best-of-N Decoding via Speculative Rejection.

2. Test-Time Preference Optimization: On-the-Fly Alignment via Iterative Textual Feedback.

3. TreeBoN: Enhancing Inference-Time Alignment with Speculative Tree-Search and Best-of-N Sampling.

4. Args: Alignment as reward-guided search.

5. Inference-time language model alignment via integrated value guidance.

---

> ### Author Response · Authors · 2025-11-17
>
> Thank you for your comments.
>
> >The method can be viewed as a length-normalized, RM-guided extension of BoN using MCMC proposals from QUEST ...
>
> **We are confused by the reviewer's comment and would appreciate clarification.**  What does the reviewer mean by "extension of BoN" and "reparameterization of BoN/MBR with additional tuning"?  Our method does not involve any tuning; it is purely a test-time inference method that does not update parameters.
>
> >In addition, since QALIGN generates samples sequentially, it forgoes the parallel efficiency of BoN, which raises questions about its overall computational practicality.
>
> We agree that it is important to understand whether these inference budgets are realistic. Let the response length be $N$. A markov chain of length $32$ produces
> $$
> N + 31 \cdot \frac{N}{2}
> = \frac{33}{2}N
> \approx 16.5\,N,
> $$
> tokens (Appendix B). For GSM8K solutions where $90\\%$ of the responses of LLama3-8B-Instruct are bellow $377$ tokens, this corresponds to roughly $6{,}220$ generated tokens per query. This compute cost is within the normal operating regime for modern reasoning LLMs. As one concrete reference point, the Qwen/Qwen3 model card (https://huggingface.co/Qwen/Qwen3-32B) explicitly recommends reserving $32{,}768$ **sequential** output tokens (plus approximately $8{,}000$ prompt tokens) for short-text scenarios, which is roughly an order of magnitude larger than the token budget used by the 32-step QALIGN run above.
>
> Thus, the compute level at which QALIGN  surpasses both  DPO@1 and  BoN up to $n=1024$ (with 32 QAlign steps as  in Figure~1) already lies inside routinely supported output-length budgets for complex reasoning tasks.
>
> >The evaluation focuses on a single policy model (LLAMA-3.1-8B-Instruct) and one reward model (TÜLU3-8B-RM), leaving it uncertain whether the observed improvements would generalize across different architectures or RM designs.
>
> As stated in the paper (lines 091, 086--087, and 368--369), we use our method with **two** language models, Tulu3-SFT-8b and Llama3-8b-instruct, and **two** reward models one based on Llama3-8b-instruct that we trained on GSM8k training data, and another, Tulu3-RM-8b provided from the Tulu3 paper.
>
> Following the reviewer’s suggestion (WTu9), we ran **new architecture-mismatched experiments** using a Llama-3.1-8B base LM and a **third** reward model (**Qwen2.5-1.5B-Instruct RM**) trained on GSM8K training data, which we encourage you to check (below WTu9 review).
>
> > The main comparison in Table 1 uses N = 1024 samples for both BoN and QALIGN, which is considerably higher than typical inference budgets (e.g., N = 16–32). Results under more practical compute settings would strengthen the study.
>
> Figure 1 from the paper reports average performance as we change the number of samples $N$ for both BoN and QALIGN. **Before $32$ samples QAlign passes BoN.**
>
> >Furthermore, the FLOPs-adjusted comparison with DPO is not fully equitable, as training FLOPs are a one-time cost whereas inference FLOPs recur for each query.   / Can the authors offer more details regarding FLOPs computation for inference-
> time and training-time methods
>
> As stated in at lines 69--75 and lines 375--377), the costs of
> training the SFT or DPO models are **not** accounted for in any of the plots. For reference, training TULU3-8B-DPO from TULU3-8B-SFT on 271k preference pairs requires approximately $1.75 \times 10^{19}$ FLOPs (lines 75 and 377), and this entire cost is excluded from our figures. This means our comparisons are extremely generous to those methods. Our focus is only on
> **inference compute**, i.e., the compute used to answer queries **after** training.
>
> Both  TULU3-8B-SFT (using several test-time methods) and TULU3-8B-DPO via **majority vote (MV) are** scaled with inference-time compute, which is explicitly stated in the manuscript (lines 69--75 and lines 375--377).
>
> > A natural baseline, sampling multiple random indices and running BoN over them, is not tested, making it unclear whether MCMC brings tangible benefit.
>
> **We are confused by the reviewer's proposed algorithm.** Over which sentence should we sample indices?
>
> > How do we compute  inference FLOPS ?
>
> Following Kaplan et al. (2020), we approximate that an LM with  $P$ parameters performs roughly $2P$ FLOPs per generated token during inference. Thus, generating $T$ tokens requires approximately $\text{FLOPs}\approx2PT$.
>
> This approximation is widely used in the literature on scaling and inference cost, including Kaplan et al. (2020), Hoffmann et al. (2022), and Sardana et al. (2024). In our FLOPs computation, we explicitly count the actual number of generated tokens per method and multiply by this factor (Appendix B).
>
> Methods that require reward-model evaluation incur an additional cost. Because in our setup, the reward model has the same size as the generative LM, reward evaluation roughly doubles the per-token FLOPs.
>
> We will expand these points in Appendix B.

---

> > ### Comment · Reviewer_J4U9 · 2025-11-27
> >
> > Thank you for the clarification and for conducting additional experiments.
> >
> > However, my main concerns remain. Conceptually, QALIGN appears closely related to BoN and MBR, i.e., essentially a length-normalized, RM-guided extension that replaces independent sampling with MCMC proposals from QUEST. It is not yet clear how this represents a fundamentally new paradigm beyond existing test-time scaling schemes, especially given that it sacrifices the inherent parallelism of BoN and other parallelizable test-time search methods.
> >
> > Empirically, Table 1 remains limited. It would be more informative to (1) include results under more realistic inference budgets (e.g., N = 16 or 32) since N = 1024 is rarely feasible in practice, and (2) expand Table 1 to include additional policy–reward model combinations to better demonstrate generality. While Figure 1 provides scaling trends, readers typically rely on Table 1 for main quantitative comparisons, so showing smaller-N and cross-model results there would make the study more balanced and convincing.
> >
> > Overall, I appreciate the authors’ clarifications and additional experiments but maintain my original evaluation given the remaining concerns above.

---

> > > ### Author Response · Authors · 2025-12-03
> > >
> > > Thank you for the comments.
> > >
> > > > Conceptually, QALIGN appears closely related to BoN and MBR ...
> > >
> > > We agree QALIGN is related to BoN and weighted majority voting or more generally MBR, and we want to emphasize that **making this relationship precise is itself part of our contribution** ( Sections 3.2 - 3.3). In our work we introduce a unified view of several test-time scaling methods as (exact or approximate) procedures for sampling from a  target distribution:
> > > $$
> > > \pi(y\mid x)\propto p_{\text{LM}}(y\mid x)\exp( r(y,x)/\beta).
> > > $$
> > >
> > > *  **MBR as self-normalized importance sampling**. MBR can be interpreted as estimating expectations under $\pi$ using i.i.d. samples $y\sim p_{\text{LM}}(\cdot\mid x)$ with weights proportional to $\exp(r(x,y)/\beta)$, which is precisely self-normalized importance sampling.
> > >
> > > * **BoN is in an heuristic maximizer of $r(y,x)$**:  BoN draws $N$ independent candidates from $p_{\text{LM}}(\cdot\mid x)$, scores them with $r(x,y)$, and returns the maximizer. As $N$ increases, this rule concentrates on high-reward tail events and approaches “maximum-reward”. Moreover, under strong assumptions on the reward-score distribution, we show that BoN can be turned into an approximation of $\pi(\cdot\mid x)$ when $N$ is chosen carefully (derivation in Appendix A). To our knowledge, this explicit ($N=N(\beta)$) connection has not been articulated or used in prior work. BoN is typically treated as a heuristic with $N$ as a compute knob rather than as an approximation to a fixed target distribution.
> > >
> > > *  **QALIGN directly targets $\pi(y\mid x)$ via MCMC (Metropolis-Hastings)**, converging to a  stationary distribution rather than using i.i.d. maximum selection (BoN) or high-variance self-normalized importance sampling (MBR). We prove and empirically validate that increasing the number of steps improves the quality of our approximation rather than over-optimization.
> > >
> > >   Importantly, QALIGN **does not “pick the maximum” over samples (independent or dependent)**. It is explicitly a sampling procedure whose output distribution is the chain’s stationary distribution.
> > >
> > > >  .. especially given that it sacrifices the inherent parallelism of BoN and other parallelizable test-time search methods.
> > >
> > > While it is true a single chain is sequential, QALIGN is practically parallel via batching. i.e. When we are doing inference on many prompts we share the iterative steps across them always to maximize hardware utilization in the same way that BoN would but with half the tokens generated. So for batched requests like the synthetic generation scenarios we discussed this is not a practical concern.

---

> ### Author Response · Authors · 2025-12-03
>
> > Table 1 remains limited. It would be more informative to (1) include results under more realistic inference budgets (e.g., N = 16 or 32)
>
> We will follow the reviewer suggestion and include more budgets in Table 1.  Below we provide a preview of these results. We measure task accuracy, comparing the baselines in the paper BON, WMV, MV and our proposed QALIGN as we increase the number of candidates.
>
> ## TruthfulQA
> | Method | 32    | 128   | 256   | 512   | 1024  |
> | ------ | ----- | ----- | ----- | ----- | ----- |
> | BON    | 0.469 | 0.477 | 0.508 | 0.492 | 0.453 |
> | WMV    | 0.449 | 0.466 | 0.468 | 0.449 | 0.461 |
> | MV     | 0.447 | 0.437 | 0.436 | 0.443 | 0.437 |
> | QALIGN | 0.421 | 0.438 | 0.469 | 0.487 | 0.485 |
>
> ## MATH-500
>
> | Method | 32    | 128   | 256   | 512   | 1024  |
> | ------ | ----- | ----- | ----- | ----- | ----- |
> | MV     | 0.450 | 0.482 | 0.480 | 0.494 | 0.494 |
> | BON    | 0.356 | 0.356 | 0.344 | 0.316 | 0.316 |
> | WMV    | 0.428 | 0.492 | 0.472 | 0.508 | 0.530 |
> | QALIGN | 0.418 | 0.514 | 0.566 | 0.592 | 0.604 |
>
> ## GSM8K
>
> | Method | 32    | 128   | 256   | 512   | 1024  |
> | ------ | ----- | ----- | ----- | ----- | ----- |
> | MV     | 0.853 | 0.866 | 0.863 | 0.862 | 0.863 |
> | WMV    | 0.833 | 0.847 | 0.848 | 0.849 | 0.852 |
> | BON    | 0.768 | 0.762 | 0.746 | 0.746 | 0.743 |
> | QALIGN | 0.796 | 0.854 | 0.862 | 0.870 | 0.882 |
>
> ## MMLU Redux
>
> | Method | 32    | 128   | 256   | 512   | 1024  |
> | ------ | ----- | ----- | ----- | ----- | ----- |
> | WMV    | 0.588 | 0.614 | 0.630 | 0.622 | 0.622 |
> | MV     | 0.577 | 0.589 | 0.594 | 0.596 | 0.594 |
> | BON    | 0.562 | 0.566 | 0.579 | 0.582 | 0.571 |
> | QALIGN | 0.530 | 0.573 | 0.583 | 0.603 | 0.622 |
>
> ## IFEval
>
> | Method | 32    | 128   | 256   | 512   | 1024  |
> | ------ | ----- | ----- | ----- | ----- | ----- |
> | WMV    | 0.617 | 0.580 | 0.556 | 0.508 | 0.495 |
> | BON    | 0.664 | 0.636 | 0.612 | 0.604 | 0.593 |
> | MV     | 0.708 | 0.719 | 0.734 | 0.719 | 0.726 |
> | QALIGN | 0.710 | 0.732 | 0.745 | 0.771 | 0.776 |
>
> > expand Table 1 to include additional policy–reward model combinations
>
>
> We additionally evaluated a new policy–reward model combination: **Qwen/Qwen2.5-1.5B-Instruct** (policy) with **Skywork/Skywork-Reward-V2-Qwen3-1.7B** (reward model), on a subset of Table 1 datasets due to time constraints. We report test accuracy at N = 32, 128, 256  and (512 for qalign ) candidates.
>
> ## GSM8K
> | Method | 32    | 128   | 256   | 512 |
> | ------ | ----- | ----- | ----- |  ----- |
> | WMV    | 0.776 | 0.786 | 0.782 |    |
> | BON    | 0.765 | 0.776 | 0.779 |      |
> | Qalign | 0.727 | 0.763 | 0.778 |   0.791|
>
>
> ## MATH-500
> | Method | 32    | 128   | 256   | 512 |
> | ------ | ----- | ----- | ----- | ----- |
> | WMV    | 0.568 | 0.608 | 0.624 |      |
> | BON    | 0.538 | 0.548 | 0.584 |    |
> | Qalign | 0.540 | 0.596 | 0.632 |  0.652 |

---

### Official Review · Reviewer_YAd4 · 2025-10-28

**Soundness:** 2
**Presentation:** 2
**Contribution:** 2
**Rating:** 4
**Confidence:** 4

**Summary:**

This paper introduces a test-time alignment method for LLMs that improves performance without additional training or access to model logits. It does so by leveraging Markov chain Monte Carlo sampling to better explore and sample from the optimal aligned output distribution as computation increases, avoiding over-optimization issues seen in prior reward-model-based methods. The result is more accurate and well-aligned responses across reasoning and preference benchmarks, outperforming existing test-time compute and alignment baselines.

**Strengths:**

1. The paper analyzes the language model sampling from a MCMC prospective, and the algorithm is novel.
2. The algorithm is surprisingly simple and does not require accessing the models' logits, making it suitable for any LLM (open-sourced or commercial).
3. The evaluation seems comprehensive.

**Weaknesses:**

1. There seems error in derivation. First, in equation (7), when computing the ratio, the right hand side's numerator does not seem to be consistent with the definition of the proposal distribution. Also, it is not very clear how equation (8) is obtained, especially how everything is reduced to the length ratio.

2. Unclear experiment setting: It is unclear how finetuning based method such as DPO and SFT is scaled during inference time. Also, can the author provide the number n for best-of-n? The Inference FLOPs is helpful, but number n is also commonly used.

3. Lack of experimenting closed-source model. One claimed advantage of the paper is that it does not require accessing the logic. It would be very helpful if some results on closed-source model can be provided.

4. This method requires selecting an index and complete the full response, for multiple times, which can be very expensive. Recently there are token-level reward model [1] that have shown to be efficient at guide a frozen LLM to generate aligned outputs. It would be helpful to also compare with such inference time alignment approach, which also seems to give the optimal distribution under the RL problem.

[1] GenARM: Reward Guided Generation with Autoregressive Reward Model for Test-Time Alignment, ICLR 2025.

**Questions:**

1. Is there a reason to choose the specific proposal distribution by QUEST? Is this choice optimal in any sense?

---

> ### Author Response · Authors · 2025-11-17
>
> Thank you for your comments.
>
> > There seems error in derivation. First, in equation (7), when computing the ratio, the right hand side's numerator does not seem to be consistent with the definition of the proposal distribution. Also, it is not very clear how equation (8) is obtained, especially how everything is reduced to the length ratio.
>
> **There is not an error in the derivation.** To clarify the derivation, we restate the step-by-step argument.
> Since the token sequences $y$ and $y^{t}$ share the same prefix from index $0$ up to (but not including)  index $i$, we can write
>
> $$y = (y_0, \ldots, y_{i-1}, y_{i}, \ldots, y_{N}) = (y_0^{t}, \ldots, y_{i-1}^{t}, y_{i}, \ldots, y_{N}), \qquad y^{t} = (y_0^{t}, \ldots, y_{i-1}^{t}, y_{i}^{t}, \ldots, y_{N}^{t}).$$
>
> Therefore, the proposal ratio (proposal density defined in Eq. 4) can be written as
>
> $$
> \frac{q(y^{t}\mid y,x,i)}{q(y \mid y^{t},x,i)}
> = \frac{p\big(y_{i:N}^{t} \mid y_{<i}, x\big)}
>      {p\big(y_{i:N} \mid y_{<i)}^{t}, x\big)}\frac{\mathbf{1}(y_{<i} = y_{<i}^{t})}{\mathbf{1}(y_{<i}^{t} = y_{<i})}
> $$
>
> Because the prefixes of $y$ and $y^t$ are the same, we can write $y_{<i} = y^{t}_{<i}$ and the indicator ratio becomes $1$.
> Thus,
>
> $$
> \frac{q(y^{t}\mid y,x,i)}{q(y \mid y^{t},x,i)}
> = \frac{p\big(y_{i:N}^{t} \mid y_{<i}^{t}, x\big)}
>      {p\big(y_{i:N} \mid y_{<i}^{t}, x\big)}
> $$
>
>
> We apply the same argument to the ratio of aligned posteriors:
>
> $$
> \frac{\pi_\beta\big(y\mid x\big)}{\pi_\beta\big(y^{t}\mid x\big)}=\exp\bigg(\frac{r\big(y,x\big) - r\big(y^{t},x\big)}{\beta}\bigg)\frac
> {p\big( y_{i:N} \mid y_{<i},x\big)p\big( y_{<i} \mid x\big)}
> {p\big( y_{i:N}^{t} \mid y_{<i}^{t},x\big)p\big( y_{<i}^{t} \mid x\big)}
> $$
>
> Again, since the prefix is the same, the likelihood terms for the prefix cancel, and we write the prefix as the value at time $t$.
>
> Combining the two ratios all language-model terms cancel, leaving
>
> $$
> \frac{\pi_\beta\big(y\mid x\big)}{\pi_\beta\big(y^{t}\mid x\big)}\frac{q(y^{t}\mid y,x,i)}{q(y \mid y^{t},x,i)}=
> \exp\bigg(\frac{r\big(y,x\big) - r\big(y^{t},x\big)}{\beta}\bigg)$$
>
>
> > Unclear experiment setting: It is unclear how finetuning based method such as DPO and SFT is scaled during inference time.
>
> As stated  in lines 69--75 and lines 375--377), the costs of
> training the SFT or DPO models are **not** accounted for in any of the plots; that is, the analysis is inherently generous to those methods. Our focus is on **inference compute**, i.e., the compute used to answer queries **after** training.
>
> Both TULU3-8B-SFT (using several test-time methods) and TULU3-8B-DPO via **majority vote (MV)**
> **are** scaled with inference-time compute, which is explicitly stated
> in the manuscript (lines 71--72, 375 and 395-396). Thus, **DPO is scaled with inference compute** through the MV baselines that appear directly in
> Figure~1. All of these test-time methods generate tokens using a particular model, and we use this
> to estimate inference FLOPs.
>
>
> >Also, can the author provide the number n for best-of-n? The Inference FLOPs is helpful, but number n is also commonly used.
>
> We used the sequence of $n$ values as follows:
> $$
> n \in \\{1, 2, 4, 8, 16, 32, 64, 128, 256, 512, 1024\\}.
> $$
>
> Thank you for pointing this out. In the revision we will make these values more explicit in the
> main text and add this table to the appendix summarizing all baselines.
>
>
> > Response to the token-level reward model comment
>
> As we elaborate in lines 451–-456, we use **outcome-based reward models**, which operate only on full generations. This is the most general setting for reward models and  currently the most commonly used kind of RM in practice (e.g., see Tülu-3, DeepSeek, Llama-3). Outcome-based RMs do not support token-wise guided decoding in the way assumed by the method cited by the reviewer.  While an interesting area for ongoing research, process/token-level reward models have reportedly been found to have limitations (e.g., by the DeepSeek paper), as we mention in lines 455--456, and as evidenced by their limited adoption so far.
>
> We agree that making different types of reward models and subsequently test-time alignment methods work effectively is an important research direction. We will mention this paper in the related work section together with the other process-level reward model methods we already discuss. In our experiments, we compare against the most widely used and better-performing baselines for outcome-based reward models, which we designed for because of their relatively weak assumptions/requirements.
>
>
> Grattafiori et al.,  The Llama 3 Herd of Models}, 2024.
>
> Lambert et al., Tulu 3: Pushing Frontiers in Open Language Model Post-Training, 2024.
>
> DeepSeek-AI,  DeepSeek-R1: Incentivizing Reasoning Capability in LLMs via Reinforcement Learning, 2025.

---

> > ### Author Response · Authors · 2025-11-17
> >
> > >Lack of experimenting closed-source model.
> >
> > We agree this is an interesting direction. We focused on open models because they provide fully reproducible setups and make the effect of test-time alignment easier to isolate, without interference from closed models being tuned to the standard benchmarks we use.
> >
> > However, QAlign does not require any property that is specific to open models. It only needs the ability to (i) sample candidate completions from a base LM and (ii) score them with a RM. This is exactly the interface exposed by standard commercial APIs, so the algorithm can be applied to closed-source models without modification. We will clarify this point in the revision and highlight a direction for future work.

---

> > > ### Comment · Reviewer_YAd4 · 2025-11-25
> > >
> > > thanks for the clarification on the derivation. As for the closed-source model, since this seems to be a significant claimed contribution of this model to deal with closed-source model, is it possible to provide results on this?

---

> > > > ### Author Response · Authors · 2025-11-28
> > > >
> > > > Following the reviewer’s request, we added results with the **closed-source**  **Claude Haiku 4.5 (2025-10-01)** as the base model (accessed via API) and **Qwen/Qwen2.5-Math-RM-72B** as the (math) reward model, evaluated on **AIME25** ( American Invitational Mathematics Examination 2025 - a test dataset we hope the closed model did not train on ). The **baselines are the ones  throughout the paper** (BoN, MV, WMV), and we report accuracy as a function of number of candidates (relative small maximum budge of 32). We used the same hyper-parameters as in experiments of Section 4.2.
> > > >
> > > > | Method | 1    | 2    | 4    | 8    | 16   | 32   |     |
> > > > | ------ | ---- | ---- | ---- | ---- | ---- | ---- | --- |
> > > > | QAlign | 0.30 | 0.30 | 0.33 | 0.33 | 0.33 | 0.43 |     |
> > > > | BoN    | 0.23 | 0.30 | 0.33 | 0.30 | 0.33 | 0.40 |     |
> > > > | WMV    | 0.23 | 0.30 | 0.33 | 0.23 | 0.33 | 0.37 |     |
> > > > | MV     | 0.23 | 0.23 | 0.23 | 0.27 | 0.27 | 0.37 |     |
> > > > These are **small inference-budget runs**: due to the limited closed-source query budget, we only evaluate up to 32 candidates, which is fewer than the budgets where we see the largest gains elsewhere in the paper. However, QAlign achieves best performance at 32, which is consistent with our main claim that QAlign can improve alignment at inference time. We also do not observe the typical BoN over-optimization in this regime, likely because the reward model is strong and substantially larger candidate sets are needed before BoN begins to exploit reward model artifacts.

---

### Official Review · Reviewer_PW39 · 2025-10-30

**Soundness:** 3
**Presentation:** 2
**Contribution:** 2
**Rating:** 2
**Confidence:** 2

**Summary:**

The paper *“SAMPLE, DON’T SEARCH: Rethinking Test-Time Alignment for Language Models”* proposes **QAlign**, a new test-time alignment method that leverages Markov chain Monte Carlo methods for text generation, improving efficiency–alignment trade-offs at high compute budgets, while maintaining the flexibility of inference-time alignment without modifying model parameters.

**Strengths:**

- **Strong empirical performance in high-compute regimes:** QAlign consistently outperforms other alignment methods when sufficient inference-time compute is available.
- **Solid experimental validation:** The experiments include diverse prompts and datasets, providing a fair comparison across multiple baselines and compute settings.

**Weaknesses:**

- **Compute intensity:** QAlign requires substantial inference-time compute to surpass baselines, limiting its practicality in typical deployment settings.
- **Limited theoretical justification:** Although the paper claims convergence to the “optimal aligned distribution,” this claim is not clearly seen to be proven. The paper would benefit from a clearer mathematical explanation or proof sketch illustrating how sampling asymptotically approximates the optimal aligned distribution.
- **Narrow advantage region:** In most compute-constrained regimes, traditional search-based or weighted-logit methods still outperform QAlign.

**Questions:**

1. QAlign appears to require high compute budgets to outperform existing baselines. In practice, are such inference-time budgets realistic or allowed?
2. Under what specific conditions (model size, compute budget, or sampling strategy) would you recommend using QAlign over search-based alternatives?
3. Can you provide a formal or empirical justification for the claimed convergence of QAlign to the optimal aligned distribution for each prompt?

---

> ### Author Response · Authors · 2025-11-17
>
> > Limited theoretical justification: Although the paper claims convergence to the “optimal aligned distribution,” this claim is not clearly seen to be proven. The paper would benefit from a clearer mathematical explanation or proof sketch illustrating how sampling asymptotically approximates the optimal aligned distribution.
>
> Below, we provide a clear and self-contained argument addressing the reviewer’s concern about the convergence claim. The theoretical justification requires two ingredients that we formulate as Proposition 1 and Proposition 2. There is, unfortunately, no way we can fit the full proof here. We will do our best to do a sketch here. **We emphasize, however, that the theoretical justification for the method is strong.**
>
> ## Proposition 1. (from Korbak et al. (2022a) and cited in line 111)
>
> For a fixed prompt $x$, the RLHF objective
> $$\mathcal{L}(\theta, x)={E}\_{y \sim q (y\mid x)} \big( r\_\phi (y,x) \beta \big) - D_{ \text{KL}}\big( q_\theta(y\mid x) \| p_{\text{LM}}(y\mid x) \big)$$
>
> where $q_\theta(y\mid x)$ is the learned policy, is maximized by the posterior
> $$
> p(y \mid \gamma=1,x)
> = \frac{1}{Z_\beta(x)} p_\text{LM}(y \mid x)\exp \left({r(y, x)}/{\beta} \right),
> $$
> which we denote by $\pi_\beta(y\mid x)$. Following the definitions in lines 109--120, $\gamma=1$ denotes a “preferred’’ response under the human preference model, and $Z_\beta(x)$ is the normalizing constant.
>
> **Proof Sketch**
>
> To prove this, we follow the standard variational inference derivation starting from the log evidence $\log p(\gamma=1\mid x)$. We write the evidence decomposition as
>
> $$
> \log p(\gamma=1\mid x)
> = \mathbb{E}\_{y \sim q\_\theta(y\mid x)} \bigg( \log  \frac{p(y,\gamma=1\mid x)}{q\_\theta(y\mid x)} \bigg) -D_{\text{KL}}\big(q\_\theta(y\mid x) \| p\_\text{LM}(y\mid x)\big)+ D_{\text{KL}}\big(q\_\theta(y\mid x) \| p(y\mid \gamma=1,x)\big)
> $$
>
> We now plug in the preference model (line 115)
> $$
> p(\gamma=1\mid y,x)
> = \exp \big( (r(y,x) - \max_y r(y,x))/\beta \big),
> $$
> and note that both $\max_{y} r(y,x)$ and $\log p(\gamma=1\mid x)$ are constants that depend only on $x$. If we now rearrange it gives:
>
> $$
> c - \mathcal{L}(\theta,x) = D_{\text{KL}}\big(q\_\theta(y\mid x) \| p(y\mid \gamma=1,x)\big)
> $$
>
> Minimizing the KL divergence is equivalent to maximizing the RLHF objective $\mathcal{L}(\theta,x)$. Therefore, the maximizer in $q_\theta$ is $ q_\theta(y\mid x) = p(y\mid \gamma=1,x)$ i.e., the posterior distribution from Eq.1.
>
> QED, this shows that $\pi_\beta(y\mid x)$ is the optimal policy for a single prompt.
>
>
> ## Proposition 2.
>
> Let $\pi_\beta(y\mid x)$ be the target distribution from Proposition 1, and consider the Metropolis–Hastings chain used by QALIGN with target $\pi_\beta$ and the QUEST proposal kernel $q(y \mid y^t,x)$. Then the distribution of $y^t$ converges in distribution to $\pi_\beta(y\mid x)$ as $t \to \infty$.
>
> **Proof Sketch**
>
>  The theory comes from the standard Metropolis--Hastings (MH) framework (**line 182**; Hastings, 1970) together with the properties of the proposal distribution introduced in QUEST (**line 177**; Faria et al., 2024). Once we specify (i) the target distribution $\pi_\beta(y\mid x)$ obtained in Proposition 1 and (ii) a valid proposal kernel, the MH theorem ensures that the resulting Markov chain converges to $\pi_\beta$ under conditions established by (Faria et al., 2024).
>
> The central requirement is that the MH transition kernel satisfies **detailed balance** with respect to~$\pi_\beta$, which, with the standard MH accept-reject rule in Eq 5, Hastings (1970) shows that detailed balance holds automatically for any proposal $q(y\mid y^t,x)$.
>
> The second requirement is that the chain is **irreducible** and **aperiodic**. QUEST (line 177; Faria et al., 2024) proves that the proposal distribution defined using LLM-based local edits is irreducible and aperiodic over the support of $\pi_\beta$.
>
> What we show in our methods section is precisely how we instantiate the MH accept--reject ratio using (i) the target distribution $\pi_\beta$ derived in Proposition 1 and (ii) the QUEST proposal from Faria et al.\ (2024). Once these are plugged into the MH update, convergence to $\pi_\beta$ follows directly from the classical MH theory.
>
> **We will add these full proofs to the Appendix for completeness.  We emphasize again, however, that the theoretical justification for the method is strong.**

---

> ### Author Response · Authors · 2025-11-17
>
> Thank you for your comments.
>
> > QAlign appears to require high compute budgets to outperform existing baselines. In practice, are such inference-time budgets realistic or allowed?
>
> We agree that it is important to understand whether the inference budgets at which QALIGN excels are realistic. To translate our setup into token usage, let the response length be $N$. A markov chain of length $32$ produces
>
> $$
> N + 31 \cdot \frac{N}{2}
> = \frac{33}{2}N
> \approx 16.5\,N,
> $$
> tokens (Appendix B). For GSM8K solutions where $90\\%$ of the responses of LLama3-8B-Instruct are bellow $377$ tokens, this corresponds to roughly $6{,}220$ generated tokens per query. This compute cost is within the normal operating regime for modern reasoning LLMs. As one concrete reference point, the Qwen/Qwen3 model card ( https://huggingface.co/Qwen/Qwen3-32B) explicitly recommends reserving $32{,}768$ output tokens (plus approximately $8{,}000$ prompt tokens) for short-text scenarios, which is roughly an order of magnitude larger than the token budget used by the 32-step QALIGN run above.
>
> Thus, the compute level at which QALIGN substantially surpasses both  DPO@1 and  BoN up to $n=1024$ (i.e. with 32 QAlign steps as observed in Figure~1) already lies inside routinely supported output-length budgets for complex reasoning tasks. Moreover, for users who are already in the regime of tens or hundreds of generations per query (as in  BoN or self-consistency/majority vote), QALIGN does not require a new or unrealistic inference-time regime; it uses the existing budget more effectively.
>
> >Under what specific conditions (model size, compute budget, or sampling strategy) would you recommend using QAlign over search-based alternatives?
>
> We recommend QALIGN in settings where users already operate in a moderate-to-high inference-compute regime and care about alignment quality: for example, (i) high-stakes queries where tens of samples per prompt are acceptable, (ii) offline synthetic data generation for post-training, and more generally deployments that already use sampling-intensive methods such as BoN or self-consistency. In these regimes, QALIGN provides better test-time alignment.
>
> Figure~1 shows that QALIGN begins to outperform both  DPO@1 (which itself required substantial **training** compute that is not contained in the plot) and all BoN baselines after roughly **32 QALIGN steps**. This is the regime we highlight in our recommendations.  BoN is by far the most commonly used test-time alignment technique and the one we primarily baseline against.
>
> Popular recent work also uses comparable sampling budgets for synthetic data generation for post-training (64 to 128 samples), such as Llama3 and TULU3. From our study, we can argue that replacing BoN with QALIGN in these inference setups would improve the quality of the generated data and, as a consequence, the quality of the post-trained model.
>
> Grattafiori et al.,  The Llama 3 Herd of Models, 2024.
>
> Lambert et al.,  Tulu 3: Pushing Frontiers in Open Language Model Post-Training, 2024.
>
> We will add these arguments to the discussion in the document.
>
> > Is there a reason to choose the specific proposal distribution by QUEST? Is this choice optimal in any sense?
>
> The choice of the QUEST proposal is not about optimality but about simplicity and applicability. As we note in **lines 225–227**, with the QUEST proposal and this target distribution, the accept–reject rule reduces to the exponential reward-difference form, and we do not need to track two sets of logits or compute the full Metropolis–Hastings ratio. If we used any other proposal distribution, we would immediately lose this simplicity and would need to compute the general MH acceptance ratio involving base-model and proposal log-probabilities. We believe this simplicity will make the method more widely adopted.

---

> ### Comment · Reviewer_PW39 · 2025-11-21
>
> The authors have answered by concerns. Given their promises of addressing them in the final draft, I shall raise my score.

---

### Official Review · Reviewer_WTu9 · 2025-11-01

**Soundness:** 2
**Presentation:** 3
**Contribution:** 3
**Rating:** 6
**Confidence:** 3

**Summary:**

This paper introduces QALIGN, a novel test-time alignment method designed to address the over-optimization problem where existing methods, like Best-of-n (BoN), degrade as compute scales due to imperfect reward models (RMs). Instead of maximizing the RM, QALIGN uses a Markov chain Monte Carlo (MCMC) sampling approach, adapted from QUEST, to draw samples from the optimal aligned posterior distribution for a given prompt. This method notably requires no model finetuning or logit access, only the ability to sample from the base LM and query an RM. The final answer is selected from the resulting samples using Minimum Bayes Risk (MBR), which amounts to majority voting for tasks with discrete answers. The authors empirically demonstrate that QALIGN consistently improves with increased computation, outperforming BoN, majority voting (MV), and weighted majority voting (WMV) , and even surpasses the performance of the finetuned DPO model on a diverse suite of benchmarks when given a comparable inference budget.

**Strengths:**

The paper's primary strength lies in its proposal of QALIGN, a novel and practical test-time alignment method that directly addresses the critical over-optimization problem where methods like Best-of-n (BoN) see performance degrade with increased compute. The method is well-motivated, and its technical approach—using MCMC sampling to approximate the optimal aligned posterior distribution—is elegant. The key advantage, which is well-supported by experiments, is that QALIGN's performance consistently improves as the compute budget scales, allowing it to avoid the performance degradation that plagues other methods. The empirical evaluation is strong, demonstrating that QALIGN not only outperforms other test-time methods (BoN, MV, WMV) but also surpasses the performance of a fully finetuned DPO model across a diverse suite of benchmarks when given a comparable inference budget.

**Weaknesses:**

**Scope of Empirical Evaluation is Limited to "Matched" Model Pairs**: The paper does not test the robustness of QALIGN when the base Language Model (LM) and the Reward Model (RM) are "mismatched." In the Task-Specific Tuning experiments (Sec 4.1), the base LM is LLAMA-3.1-8B-INSTRUCT, and the RM used is a custom model finetuned from that same LLAMA-3.1-8B-INSTRUCT model. In the General Alignment experiments (Sec 4.2), the base LM is TÜLU3-8B-SFT, which is paired with the TÜLU3-8B-RM. These models are from the same family and were explicitly chosen for their close relationship to allow for a fair comparison with the TÜLU3-8B-DPO model.

**Questions:**

1. **Robustness to Mismatch**: Could the authors please comment on the robustness of QALIGN to "mismatched" model pairs? For instance, how would the method perform if the TÜLU3-8B-RM were used to align a different base model, such as a Llama 3.1 or Qwen 2.5 model?

2. **RM Dependence**: How dependent is QALIGN's success on the RM being trained (or finetuned) on the specific output distribution of the base LM? Is it possible that the MCMC sampling process becomes inefficient or unstable if the RM's preferred distribution is too far from the base LM's proposals?

---

> ### Author Response · Authors · 2025-11-17
>
> Thank you for your positive and insightful comments. We address your questions below:
>
>    > Scope of Empirical Evaluation is Limited to "Matched" Model Pairs: The paper does not test the robustness of QALIGN when the base Language Model (LM) and the Reward Model (RM) are "mismatched."
>
> Thank you for raising the important distinction between **architecture mismatch** and **data mismatch**. We address both below.
>
> In Section 4.2, despite the name similarity, the  TÜLU3-8B-RM is, in fact **mismatched**  to the TÜLU3-8B-SFT base LM in terms of data distribution. As described in the TÜLU3 paper, this RM is trained on preference data from **22 heterogeneous model families** (Yi, Gemma, InternLM2.5, Mistral, Qwen2.5, Falcon, multiple Llama versions, etc.).
> This means the RM represents a mixture-of-models distribution that is substantially different from the base LM’s output distribution.
> We will explicitly incorporate this clarification in the revision.
>
> QALIGN is stable under this strong  data mismatch, while BoN quickly degrades (lines 423–-425), directly supporting our over-optimization analysis.
>
> Following the reviewer’s suggestion, we ran **new architecture-mismatched experiments** using a  Llama-3.1-8B base LM and a **Qwen2.5-1.5B**-Instruct RM trained on GSM8K training data. We evaluate on GSM8K-Symbolic and GSM8K test sets for up to 128 steps, following the same setup as in Section 4.1. The following table reports answer accuracy ( higher is better) as a function of the number of solutions generated for each problem using BoN or QAlign. Different columns refer to different number of generated solutions.
>
> **GSM8k-Symbolic:**
> | Method     | 1     | 2     | 4     | 8     | 16    | 32    | 64    | 128   |
> |-----------|-------|-------|-------|-------|-------|-------|-------|-------|
> | **BoN**   | 0.530 | 0.532 | 0.558 | 0.569 | 0.545 | 0.534 | 0.528 | 0.527 |
> | **QALIGN**| 0.566 | 0.566 | 0.623 | 0.693 | 0.740 | 0.781 | 0.811 | 0.843 |
>
> **GSM8k:**
> | Method      | 1     | 2     | 4     | 8     | 16    | 32    | 64    | 128   |
> |------------|-------|-------|-------|-------|-------|-------|-------|-------|
> | **QALIGN** | 0.763 | 0.763 | 0.799 | 0.829 | 0.841 | 0.860 | 0.882 | 0.878 |
> | **BoN**    | 0.758 | 0.765 | 0.755 | 0.739 | 0.735 | 0.728 | 0.719 | 0.706 |
>
> We observe the same qualitative trend as in Section 4.2. BoN exhibits early over-optimization. The accuracy peaks early and then degrades as the number of samples increases. In contrast, QALIGN continues to improve on both GSM8K-Symbolic and GSM8K and reaches substantially higher accuracy at $n = 128$. **In total, the experimental results show that QALIGN is robust to both data and architecture mismatch between the base LM and the RM.**
>
>
>
> **References:**
>
> Lambert et al., Tulu 3: Pushing Frontiers in Open Language Model Post-Training, 2024.
>
>
> Thank you,
>
> Authors

---

### Author Response · Authors · 2025-11-17

We thank all the reviewers for their constructive and helpful feedback.

We are glad that the reviewers found our approach to be **novel, principled, and practical test-time alignment method** that directly addresses over-optimization under imperfect reward models (WTu9, J4U9). They noted that it provides strong and scalable empirical performance (WTu9, PW39, J4U9). They stressed that the **approach is simple, elegant, and broadly applicable** (J4U9, YAd4) and, they agreed that the **experimental evaluation is thorough and robust**, spanning diverse prompts, datasets, and both task-specific and general-alignment settings (WTu9, PW39, YAd4, J4U9).

We believe that the straightforward revisions we will make in response to your feedback will significantly improve the quality and clarity of the paper. We will release the code to facilitate the reproducibility of our results upon acceptance. If we have succeeded in responding to your comments, kindly consider raising the scores. We are happy to address any more questions you might have.

Thank you,

Authors.

---

### Meta-Review · Area_Chair_oLim · 2025-12-31

**Summary:**

The paper “SAMPLE, DON’T SEARCH: Rethinking Test-Time Alignment for Language Models” proposes QAlign, a new test-time alignment method that leverages Markov chain Monte Carlo methods for text generation, improving efficiency–alignment trade-offs at high compute budgets, while maintaining the flexibility of inference-time alignment without modifying model parameters.

**Reviewer Concerns:**

1. The scope of Empirical Evaluation is Limited to "Matched" Model Pairs.
2. Limited theoretical justification: Although the paper claims convergence to the “optimal aligned distribution,” this claim is not clearly seen to be proven. The paper would benefit from a clearer mathematical explanation or proof sketch illustrating how sampling asymptotically approximates the optimal aligned distribution
3. Lack of experimentation with a closed-source model. One claimed advantage of the paper is that it does not require accessing the logic.
4. The related work section is relatively comprehensive but misses several works on inference-time alignment.

**Reviewer Scores:**

The paper received diverse ratings initially. The authors have provided the rebuttal and addressed some of the concerns. However, some of the main concerns still exist. The final decision from AC is rejection.

---

### Decision · Program_Chairs · 2026-01-26

Reject